# Short- and Long-Term Effects of Birth Weight and Neonatal Care in Pigs

**DOI:** 10.3390/ani12212936

**Published:** 2022-10-26

**Authors:** María Romero, Luis Calvo, José Ignacio Morales, Ana Isabel Rodríguez, Rosa María Escudero, Álvaro Olivares, Clemente López-Bote

**Affiliations:** 1Departamento de Producción Animal, Facultad de Veterinaria, Universidad Complutense, Avda. Puerta de Hierro s/n, 28040 Madrid, Spain; 2Copiso, Avda. de Valladolid, 105, 42005 Soria, Spain; 3Incarlopsa, Ctra. N-400, Km. 95,4, 16400 Cuenca, Spain

**Keywords:** piglet, viability, neonatal care, birth weight, meat quality

## Abstract

**Simple Summary:**

Neonatal piglet viability is decreasing as a consequence of the selection for increasing numbers of piglets born per sow per year. Several strategies have been proposed to reduce neonatal mortality, which involve all aspects of swine production. Early management intervention has proven effective to reduce piglet mortality particularly in low-birth-weight piglets, although this practice, in some cases, increases production costs. The objective of this study was to assess the effect of body weight at birth and individual neonatal care provided to piglets on preweaning mortality, and the long-term effects on growth and carcass and meat characteristics. The results of this research showed that early neonatal care may be a useful practice to reduce mortality, especially in low-birth-weight piglets. Moreover, neonatal care could affect meat quality, fat content, and fatty acid profile, thus suggesting long-term effects on metabolism.

**Abstract:**

Swine industries worldwide face a loss in profit due to high piglet mortality, particularly as a consequence of the marked increase in prolificity and low birth weight (BW) of piglets. This research studied the effect of BW and individual neonatal care provided to piglets on preweaning mortality, and the long-term effects on growth and carcass and meat characteristics. Litters from seventy-one crossbred sows (PIC 34) were included in the trial. Half of each litter did not receive any further management, and the remaining half received the pre-established management protocol of early assistance of neonatal care (NC). Along lactation, the low-BW piglets (weight equal to or less than 1.1 kg) showed a threefold higher mortality rate than piglets of higher weights (32 vs. 10%; *p* = 0.001), with mortality particularly concentrated within the first week after birth. No effect of NC treatment was observed on mortality ratio caused by crushing, but a significant effect was observed in low-BW piglets who died of starvation (*p* < 0.01). The effect of NC on growth is dependent on BW, and heavier piglets at birth benefit from NC treatment to a higher extent than low-BW piglets. Low-BW piglets showed a higher fatness (*p* = 0.003), lower lean cut yield (*p* = 0.002) in carcasses, and higher intramuscular fat (IMF) content (2.29% vs. 1.91%; *p* = 0.01) in meat. NC treatment increased the lean content in carcasses from low-BW piglets (*p* < 0.01). The monounsaturated fatty acids concentration was higher in lower-than-normal-BW piglets (48.1% vs. 47.1%; *p* = 0.002) and the opposite effect was observed for polyunsaturated fatty acids (13.6% vs. 15.7%; *p* = 0.002). NC treatment induced a higher concentration of n-7 fatty acids. In conclusion, NC treatment may be a useful practice to reduce mortality in low-BW piglets. Moreover, NC could affect carcass fatness and meat quality, thus suggesting a long-term effect on metabolism.

## 1. Introduction

Piglet viability is a major problem in the swine industry, particularly in recent years due to the marked increase in prolificity. In commercial farming, it is estimated that 15–20% of piglets die during the first days of life (mainly by crushing, starvation, and chilling) [1], a matter of growing importance in the swine industry, because it not only impairs productivity, but also seriously compromises animal welfare and the social acceptability of farming practices.

Neonatal death is particularly concentrated in low-birth-weight (BW) piglets, which is usually associated with limited energy reserves, hypothermia, anoxia, and low vitality and mobility during the first initial hours [2]. The proportion of small piglets (<1.1 kg) in litters with more than 15 piglets is above 15–20% [3,4,5,6].

Several strategies have been proposed to reduce neonatal mortality, which involve all aspects of swine production: nutritional strategies to improve fetal development, management practices during gestation, and transitional and peripartum periods to improve the performance of neonatal piglets. Additionally, management practices in early lactation including farrowing supervision and assistance to piglets, maximizing colostrum intake, cross-fostering techniques, providing nurse strategies, and providing artificial milk have been suggested [1,7]. This combination of strategies has been reported to reduce neonatal mortality, and a figure of 5–7% may be considered achievable [8]. The early management intervention of piglets and the promotion of colostrum intake have proven effective to reduce piglet mortality, particularly in low-BW piglets [9,10,11,12,13]. Nevertheless, the implementation of these management practices within production is usually low, as it requires extra labor cost, which, in many cases, does not compensate the potential benefit of reducing neonatal mortality.

On the other hand, low BW is associated with different developmental patterns, and changes in body composition and homeostasis, due to prenatal programming through epigenetic changes [14,15,16], which is associated with a high variability in carcass and meat quality in the same feedlot, which affects profitability and the production of commercial pig products [17]. Thus, gestational events (fetal programming and intrauterine growth retardation) and early neonatal nutrition (colostrum intake) have been shown to produce long-term effects in pigs, markedly affecting growth, body composition, and physiological regulation, and emphasize the importance of gestational diets and peripartum management [3,13,18,19,20,21,22,23,24,25]. Therefore, it may be speculated that neonatal intervention in piglets may produce effects beyond decreasing mortality that may justify implementation of management strategies in the commercial setting.

The aim of this study was to study the effect of BW and individual neonatal care provided to newborn piglets according to sow parity and sex on preweaning mortality, and the long-term effects on growth and carcass and meat characteristics.

## 2. Materials and Methods

### 2.1. Animals and Housing

The experiment was performed at a commercial farm of COPISO, located at Cubo de la Solana (Soria, Spain) between January and July 2020. All the animals were housed in accordance with the EU (European Directive 2008/120/EC). In compliance with European Directive 2010/63/EC Article 1 5. (f), the present study did not imply any invasive procedure or treatment to the animals. The study was reviewed and approved by the Animal Welfare Committee of the Comunidad Castilla Leon (Ref Fr/bb-2022). The sows participating in the experiment were 71 crossbred sows (Landrace × Large White, PIC 34) with parity (P) range 1–8 (P1 = 17 gilts; P2 = 4; P3= 16; P4 = 7; P5 = 7; P6 = 2; P7 = 1; P8 = 17 sows). Sows were divided into two parity classes: first parity (gilt) and second or higher parity (sow). Heterospermic inseminations were carried out with freshly collected semen from Danbred Sires. Semen was maintained under refrigeration and was used within the first 48 h after collection.

### 2.2. Farrowing, Neonatal Care, and Lactation

Along gestation, sows and gilts were kept in groups of 60 in gestation pens. All multiparous sows had farrowed in pens with farrowing crates in their previous gestations. One week prior to farrowing, gestating sows were moved to pens with farrowing crates with a fully slatted plastic floor. Two rooms existed for each farrowing system and all rooms were in the same building. The farrowing box was 260 cm long and 200 cm wide, and the dimensions of the farrowing crate were 190 cm × 80 cm.

A peripartum diet (16.5% CP, 2.15 McalEN/kg, and 3.95 g SID LYS/Mcal EN) was fed restrictedly twice a day (7.00 h and 15.00 h) at 3 kg/sow/d from gestation day 112 to farrowing until 4 days after farrowing (based on the individual farrowing date of the sow). After farrowing, diets were provided at 3.5, 4, 4.5, and 5 kg/sow/d. Starting at day 5 of lactation to weaning, a commercially available lactation diet (16.6% CP, 2.3 Mcal EN/kg, and 3.69 g SID LYS/McalEN) was provided by means of a Spotmix system (Schauer-Agrotronic, Prambachkirchen, Austria) by increasing 0.5 kg/sow/d until maximum voluntary intake. Sows had ad libitum access to drinking water. Piglets had access to nipple drinkers and were not provided with creep feeding. The daily work routine was kept consistent across the farrowing systems to reduce sources of variation.

At birth, all piglets were individually weighed and identified with electronic ear tags (MPIG-DATA, Madrid, Spain). Half of the piglets per littler did not receive any further management (control group; CT) and were returned to the same location within the crate from which they were removed, and the other half (chosen according to similar weight distribution) received the pre-established management protocol of early assistance of neonatal care (NC), which consisted of: (a) drying and massaging the whole body of the piglet with a clean and dry paper towel for approximately one minute, (b) tying the umbilical cord, (c) removing mucus and other debris from oral and nasal cavities, and (d) softly introducing an available nipple into the piglets’ mouth. Piglets’ weights equal to or less than 1.1 kg were classified as low BW [6], and the remaining were considered normal BW.

Piglets exceeding available teats were cross-fostered to other sows and were not followed along the experiment. When cross-fostering was necessary, it was carried out within 24 h after birth and exceeding piglets were removed from the litter, ensuring that the litter maintained the same proportion of low- and normal-BW piglets. Tail docking was conducted under veterinary advisement on day 4 postpartum, as well as castration of male piglets.

Piglets were individually weighed at 0, 1 d, and 7 d of life and at weaning. Piglet losses were recorded daily. Apparent causes of death (crushing, starvation/low vitality, and diarrhea) were determined by the personnel of the farm with the assistance of the veterinarian in charge. Weaning was conducted at a mean value of 25.8 ± 1.9 days, with no differences according to parity.

### 2.3. Weaning, Growing, and Fattening

Weaned piglets were maintained in groups of 24 in flat-deck pens (2.60 × 2.35 m) with a common feeder (1.5 m length) and two water bowl drinkers. Room temperature was maintained at 30 ± 2 °C for the first week of the experiment and then reduced 2 °C per week until reaching 24 °C. The pigs had free access to feed and water throughout the experiment with ad libitum feeding. Diet and handling were identical to all animals independently of previously assigned treatments. A commercially available weaning diet (until two first weeks after weaning) (18.5% CP, 2.519 Mcal EN/kg, and 5.59 g SID Lys/Mcal EN) and starter diet (three following weeks; 18% CP, 2.45 Mcal EN/kg, and 5.38 g SID LYS/Mcal EN) were provided for ad libitum intake to all piglets.

All replicate pens had a similar average BW and same proportion (50:50) of males and females. Litter origin was not considered for the assignment of pigs to replicates.

On arrival to the growing-finishing farm, pigs were housed in a naturally ventilated finishing barn in 3.1 × 3.1 m pens with 80% slatted concrete floors with 9 pigs per pen. Pigs were examined daily by experienced staff and weekly by the veterinarian in charge of the farm. Feed in pelleted form and water were offered for ad libitum intake throughout the experiment. The feeding program was common for all the pigs and consisted of diets based on cereals and soybean meal to meet or exceed the nutrient requirements of pigs [26]. Five diets were formulated, which contained 17.7, 16.9, 14.6, 12, 3, and 11.2% CP; 2.4, 2.4, 2.55, 2.55, and 2.55 kcal EN/kg; 4.5, 4.2, 3.36, 2.7, and 2.3 g SID LYS/Mcal EN, respectively, for weight ranges of approximately 20–30, 30–45, 45–60, 60–90, and 90–120 kg. All pigs were slaughtered the same day (with 170–175 d of age).

### 2.4. Carcass and Meat Quality

The pigs were fasted for 12 h and transported 300 km to a commercial abattoir (Incarlopsa, Tarancón, Cuenca) where they were allowed a 5 h lairage with availability to rest. Pigs were stunned in a 95% CO_2_ atmosphere, then slaughtered by exsanguination and scalded at 65 °C according to standard commercial procedures. Lean content of the carcass and percentage of lean and fat of the hams were measured using the Autofom classification system (Carometec Spain, S.L., Barcelona, Spain).

Carcasses were eviscerated and weighed individually. Then, the head was removed at the atlanto-occipital junction and the carcasses were suspended in the air and refrigerated at 2 °C, with a 90% relative humidity and an air speed in the chilling room of 1 m/s for 2 h. Subcutaneous fat depth, between the third and the fourth last ribs 6–8 cm away from the body midline at the last rib level (P2) and on external ham, was measured in each carcass. Muscle ultimate pH (pHu) was measured by means of a K21 pH-meter (NWK Thien, Landsberg, Germany).

Carcasses cutting was performed in accordance with the simplified European Community reference method [27]. Hams and loins were collected and kept in the chilled room at 4 °C for 24 h and weighed.

Approximately 200 g samples of SM, BF, and LD (at the level of the last rib) were collected and located in individual plastic bags. Electric conductivity (LFStar conductivity meter, Mattahäus Ingenieurbüro, Klausa, Denmark, drip loss [28], moisture, and intramuscular fat (Foss 6500 Spectrophotometer; Foss Analytical, Barcelona, Spain) were determined.

Color measurement was performed by means of a Chroma Meter (CM-2002, Minolta, Camera, Osaka, Japan) calibrated against a white tile in accordance with the manufacturer’s recommendations (CIE, 1976). Conditions for measurement were D65 illuminant, observer 2 in SCI mode, and 1 cm aperture. The average of five random readings was used to measure lowness (L), redness (a), and yellowness (b).

The remaining meat samples were vacuum-packaged and frozen at −20 °C until analyses of intramuscular fat and fatty acid composition. Lipids from muscle samples were extracted and methylated using the procedure described by Segura and Lopez-Bote [29]. Lyophilized samples were accurately weighed in a safe-lock micro-test-tube, homogenized in 1.5 mL of dichloromethane-methanol (8:2), and mixed in a mixer mill (MM400, Retsch technology, Haan, Germany). The final biphasic system was separated by centrifugation (8 min at 10,000 rpm) and the collected solvent was evaporated under a nitrogen stream. The lipid content was gravimetrically determined. Lipid fractions were subjected to esterification by heating the obtained fractions at 80 °C for 1 h in 3 mL of methanol: toluene: H2SO4 (88:10:2 by volume).

### 2.5. Statistical Analysis

Statistical analysis was carried out using the software SAS 9.4 (SAS Institute Inc., Cary, NC, USA). A *p*-value <0.05 was accepted as significant. Mortality data were analyzed by the chi-square procedure and differences in weight and weight gain, as well as carcass fatness in relation to BW, were analyzed by using linear regression. Differences between slopes in regression fitting were analyzed by Student’s t-test. Carcass and compositional data were tested for normal distribution using histograms, and a Shapiro–Wilk test was used to determine a suitable statistical model for the evaluation. Data were analyzed following a completely randomized design using the general linear model (GLM) procedure. Fixed effects were parity, birth weight, neonatal care treatment (T), and sex (S), and the individual sow was the random effect in the statistical model. No effect of litter size was observed and, therefore, it was not included in the model. Data were presented as the mean of each group and the standard deviation (SD) together with significance levels (*p* value) of the main effects and interactions. Differences between means were considered statistically significant at *p* < 0.05. For carcass and meat quality attributes, carcass weight was used as the covariable and data are presented as LS means.

## 3. Results

### 3.1. Mortality

The effect of parity, neonatal care treatment, and birth weight on preweaning mortality is shown in Table 1. Total preweaning mortality was markedly affected by BW, and the low-BW piglets showed a three-fold higher mortality rate than piglets of higher weights (32 vs. 10%) (*p* = 0.001). Mortality was specially concentrated within the first week of life, particularly in low-BW piglets. No effect of parity or NC treatment was observed.

No effect of P or NC treatment of BW was observed on mortality ratio caused by crushing, but a significant effect of BW was observed on piglets who died of starvation or diarrhea and of T on starvation (Table 1). The effect of NC treatment on mortality caused by starvation was evident in piglets from low BW (*p* < 0.01) but not in those of higher BW (*p* > 0.1).

### 3.2. Weight and Gains

Regression analysis was used to study the effect of BW and NC on different stages of growth (weight gain). Best fitting was achieved with linear regression. Comparisons between slopes were used to evidence the effect of NC. Regression equations are shown in Table 2. Results are also shown in Figure 1.

The effect of BW and NC treatment of slaughter weight is shown in Figure 2 (left). In agreement with weight gain results, final weight is markedly affected by BW and also by NC (NC slaughter weight (kg) = 105.5 (±2.94) + 15.2 (±2.02) × BW (kg); R^2^ = 0.16; *p* < 0.0001; CT slaughter weight (kg) = 103.4 (±2.8) + 17.2 (±1.97) × BW; R^2^ = 0.24; *p* < 0.0001). As expected, in this case, slopes were different (*p* < 0.01) between experimental groups.

### 3.3. Carcass Composition and Fatness

The effect of P, NC treatment, BW, and sex on carcass and splitting yield is shown in Table 3. A marked effect of BW and sex on carcass leanness and loin and ham weights was observed in all cases. Carcasses from low-BW piglets showed a higher fatness and lower lean cut yield. To further study the effect of BW and NC on carcass leanness, we also carried out regression analysis, which produced the following equations: % Lean (CT Group) = 58.91 (±0.711) + 0.74 (±0.489) × BW; R^2^ = 0.16; *p* < 0.0001; % Lean (NC group) = 59.77 (±0.6) + 0.27 (±0.046) × BW; R^2^ = 0.24; *p* < 0.0001. In this case also, the slope was different (*p* < 0.01).

### 3.4. Meat Composition

The effect of P, T, BW, and sex on selected meat quality characteristics is shown in Table 4. A significant effect of parity was observed for pHu (*p* = 0.044), intramuscular fat in semimembranosus muscle (*p* = 0.005), and color a value. Moreover, a tendency for T × BW interaction was observed for carcass evaporative weight loss and soluble protein, and a significant effect of this same interaction was seen for electric conductivity (Figure 3A,B).

Fatty acid composition is shown in Table 5. A major effect of BW and sex was observed, which involved most fatty acids. Monounsaturated fatty acid concentration was higher in low- than in normal-BW piglets (*p* = 0.002) and in castrated than in males (*p* = 0.05), while the opposite was observed for polyunsaturated fatty acids (*p* = 0.002). A higher concentration of n-7 fatty acids was observed in neonatal piglets subjected to NC treatment than in those of the control group.

## 4. Discussion

### 4.1. Mortality

Along lactation, the low-BW piglets showed a threefold higher mortality rate than piglets of higher weights, with mortality being particularly concentrated within the first week post-birth. Our results are in agreement with previously reported data by other authors where piglet BW was generally considered the main cause for neonatal mortality [30]. Thus, according to Feldpausch et al. [6], threshold BW is approximately 1.11 kg, with the pre-weaning mortality of piglets below and over this figure being 34.4% and 8.2%, respectively.

Low-BW piglets have lower body reserves, including glycogen [31] and subcutaneous fat [32]. Limited fat insulation and catabolic heat production together with a wet surface after farrowing make this animal particularly vulnerable to hypothermia. Hypothermia makes piglets liable to reduced vigor and colostrum intake, and all these aspects in combination are a leading cause of mortality [33].

No effect of parity or NC treatment was observed. Analysis of the cause of death revealed that P, NC, and BW did not influence the proportion of mortality caused by crushing, whereas BW affected piglets’ mortality by starvation or diarrhea and NC treatment decreased mortality caused by starvation. Moreover, a closer look at the effect of T provided to piglets from different body weights showed a reduction in mortality when provided to low-BW piglets, but a lack of effectiveness when treatment was performed on higher-BW piglets.

Christison et al. [34] observed that survivability increased when piglets are dried and rubbed immediately after farrowing, and Pol et al. [35] observed an increase in rectal temperature. Moreover, Andersen et al. [36] observed that assisting piglets in finding a teat reduced mortality. Vasdal et al. [37] also reported that the combination of early drying and manual approaching at the udder reduced mortality. In our experiment, we observed that this effect is produced only in low-BW piglets, which reinforces the idea that the benefit of neonatal treatment on reducing mortality may be associated with stimulating circulation and reducing hypothermia. Pedersen et al. [38] observed that rectal temperature after birth in piglets not subjected to neonatal care was positively correlated with BW, which suggests that the beneficial effect of drying at birth on maintaining homeothermy is more effective in low- than in normal-BW piglets. According to the results reported in this manuscript, the extra cost tied to the implementation of neonatal care practices in the commercial setting could be minimized if effort is concentrated in low-BW piglets below approximately 1.1 kg.

### 4.2. Weight and Gains

In agreement with previous studies [3,5,7,14], a significant effect of BW was observed on weights and gains throughout the period of study (Figure 1 and Figure 2). It is interesting to note that in all cases, slopes in the response were higher for NC piglets, thus indicating that heavier piglets at birth benefit from NC treatment to a higher extent than low-BW piglets. Moreover, during the initial steps of growth (Figure 1a,b), NC treatment negatively affected the weight gain of low-BW piglets, while in later stages of growth (post-weaning and growing-finishing phases), low-BW pigs subjected to NC showed similar gains that those of the control group (Figure 1c,d). A possible explanation may be that NC treatment provided to low-BW pig decreases mortality and, therefore, increases the number of living pigs with limited growing capability in this group, especially during the initial stages of growth. In this sense, other authors have observed that low-BW piglets showed a reduced capacity of compensatory growth [39,40]. However, this observation requires further research as it may impact the outcomes of implementing recovery strategies applied to newborn piglets, which apparently may never reach full biological potential growth.

### 4.3. Carcass Composition and Fatness

In the control group, carcasses from low-BW piglets showed a higher fatness and lower lean cut yield, which agrees with previous findings [14,41]. Neonatal treatment led to an enhancement of lean content in carcasses from low-BW piglets. It is generally recognized that low-BW piglets (particularly those considered intrauterine-growth-retarded, IUGR) are more obesogenic [42], but the effect of neonatal care on reducing the fatness of those low-BW piglets is a novel finding, which may open new interesting scientific and technological insights. It should be noted that this experiment was carried out under productive farms conditions, and some aspects of importance such as individual sire effect were not controlled and analyzed (heterospermic insemination was used). This may lead to a weakness in assessing carcass characteristics.

Intramuscular fat concentration was higher in low-BW piglets in the three muscles under study and the moisture content was lower. Vazquez-Gómez et al. [43] also observed a higher concentration of intramuscular fat in low than in normal BW in the gluteus medius muscle (4.0 vs. 3.3%; *p* < 0.05), and these authors did not observe any effect in the LD muscle. These results are of importance in some productive circumstances, where intramuscular fat content is highly appreciated. The higher fatness (including intramuscular fat) in meat from low-BW piglets may be of particular importance in meat aimed to produce quality meat products, such as dry-cured ham. No effect of NC treatment was observed in carcass fatness or intramuscular fat content, but a higher intramuscular fat and lower moisture content were observed in castrated males than in females, which agrees with previous findings [44,45].

### 4.4. Meat Composition

Neonatal care treatment showed a tendency for moisture content in the semimembranosus muscle, and biceps femoris and semimembranosus color a value. A tendency for lower drip loss was also observed in low-BW piglets at birth. Moreover, a tendency for T × BW interaction was observed for carcass evaporative weight loss and soluble protein, and a significant effect of this same interaction was seen for electric conductivity. Although the effect is difficult to explain, because there is no current research in this specific area, it is interesting to note that in all these cases, the effect follows a similar pattern, suggesting a different effect of NC treatment on meat quality attributes related to pH drop, and subsequent protein denaturation and exudative loss. In all cases, NC provided to low-BW piglets seemed to produce a negative effect, which was not observed (or even follow a different trend) in normal-BW piglets.

A marked effect of BW was observed on fatty acid composition. Monounsaturated fatty acid concentration was higher in low- than in normal-BW pigs and in castrated than in males, while the opposite was observed for polyunsaturated fatty acids. Vazquez-Gómez et al. [43] also observed higher MUFA concentrations in low- than in normal-BW pigs at slaughter in polar lipids in an experiment carried out with local fatty pigs. As opposed to neutral (mostly triglycerides stored in adipocytes), polar (membrane) lipids play important metabolic and regulatory roles. In fatty pigs, with an intramuscular fat concentration over 7%, the predominant fraction of lipids is neutral, while in lean-improved swine genotypes, the proportion of polar lipid is proportionally of higher importance, as the membrane composition is quantitatively more stable than storage lipids.

A higher concentration of C16:1 n-7 (*p* = 0.05) and a tendency for higher concentrations of C18:1 n-7 fatty acids were observed in intramuscular lipids from pigs subjected to NC treatment than in the control groups. To the best of our knowledge, this is a novel finding that requires further research, as it may have possible implications. Palmitoleic C16:1 n-7 is a monounsaturated fatty acid produced virtually exclusively through desaturation of C16:0 via stearoyl-CoA desaturase-1 (SCD), thus reflecting the overall desaturate activity of fats. SCD is the rate-limiting enzyme for the biosynthesis of most MUFA present in tissues. In pigs, a positive correlation has been reported between IMF content and the concentration of MUFA within intramuscular fat [46], both aspects considered of importance when talking about pork quality due to technological and nutritional reported benefits (LB). Nutritional and genetic strategies to increase MUFA concentration are common in the swine industry. It is shown that pigs carrying the T allele in the promoter region of the 18 stearoyl-CoA desaturase gene lead to produced dry-cured hams within creased C16:1, C18:1n-9, C18:1n-7, and MUFA.

Alternatively, C16:1 n-7 may be considered a lipokine able to regulate different metabolic processes such as insulin sensitivity in muscle, prevention of endoplasmic reticulum oxidative stress, and lipogenic activity in white adipocytes [47]. It may be used as a metabolic marker for obesity and fatness in humans [48], but when administered exogenously into bovine adipocytes in vitro, it downregulates de novo lipogenesis and upregulates fatty acid beta-oxidation to direct fatty acids toward energy production [49].

Fatty acid composition was markedly affected by sex, in agreement with previous studies [44,45]. In summary, castrated males had a higher MUFA and lower SFA and PUFA than females.

## 5. Conclusions

A marked effect of BW exists on piglet mortality. NC treatment to low-BW piglets may be a useful practice to reduce mortality caused by starvation. The effect of NC on growth is dependent on BW, and heavier piglets at birth benefit from NC treatment to a higher extent than low-BW piglets. Carcasses from low-BW piglets are fatter and intramuscular fat is higher, but NC treatment increases carcass leanness in low-BW pigs. NC increases the concentration of some monounsaturated fatty acids and shows a tendency for some attributes related to post-mortem metabolism. The long-term effect of NC strategies provided to newborn piglets may justify the use of these practices that have proven to be effective in reducing neonatal mortality but are frequently not used in common farm practice, due to the extra labor cost, which not always pays for it if only short-term effects are considered.

## Figures and Tables

**Figure 1 animals-12-02936-f001:**
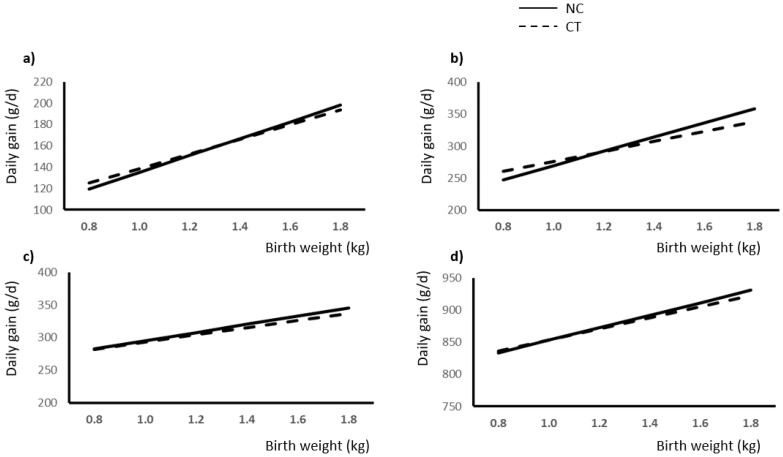
Relationship between birth weight (kg) and daily gain (g/d) as a consequence of the presence (NC) or absence (CT) of neonatal care during: (**a**) first week of lactation; (**b**) late lactation (>7 days until weaning); (**c**) weaned piglet (5 weeks after weaning); (**d**) growing-finishing period.

**Figure 2 animals-12-02936-f002:**
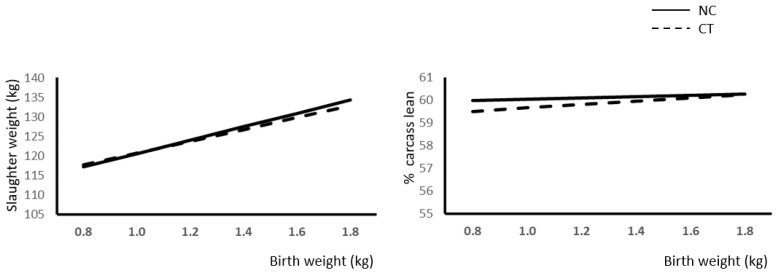
Relationship between birth weight and slaughter weight (**left**) and % carcass lean (**right**) as a consequence of the presence (NC) or absence (CT) of neonatal cares.

**Figure 3 animals-12-02936-f003:**
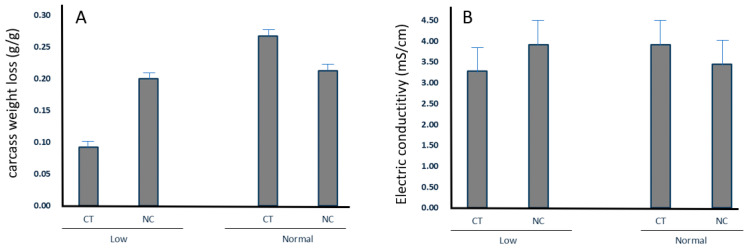
Interaction effect between neonatal care treatment and birth weight on carcass weight loss (**A**) and meat electric conductivity (**B**).

**Table 1 animals-12-02936-t001:** Effect of parity (P), neonatal care treatment (T), and birth weight (BW) on preweaning mortality and causes of death of piglets and chi-square comparison in ratio of mortality (CT: control; NC: neonatal care; low BW: weight equal to or less than 1.1 kg; normal BW: weight over 1.1 kg).

		Parity (P)		Treatment (T)		Birth Weight (BW)		*p*-Value		
		Gilt	Sow	CT	NC	Low	Normal	P	T	BW
Mortality										
	First week	0.13	0.13	0.151	0.117	0.21	0.05	0.661	0.610	0.001
	Late lactation	0.08	0.09	0.067	0.095	0.11	0.05	0.932	0.080	0.001
	Whole lactation	0.21	0.22	0.213	0.209	0.32	0.10	0.900	0.087	0.001
Cause of death										
	Crushing	0.08	0.06	0.06	0.08	0.08	0.06	0.186	0.376	0.403
	Starvation	0.04	0.07	0.09	0.02	0.10	0.01	0.297	0.032	0.001
	Diarrhea	0.01	0.00	0.01	0.00	0.01	0.00	0.370	0.158	0.029

**Table 2 animals-12-02936-t002:** Relationship between birth weight (kg) and daily gain (g/d) as a consequence of the presence (NC) or absence (CT) of neonatal care during: first week of lactation; late lactation (>7 days until weaning); weaned piglet (5 weeks after weaning); growing-finishing period.

First week of lactation:
(CT) g/d = 70.3 (±12.4) + 68.4 (±8.5) × BW (kg); R^2^ = 0.16; *p* < 0.0001
(NC) g/d = 56.8 (±11.4) + 78.4 (±8.0) × BW (kg); R^2^ = 0.24; *p* < 0.0001
Late lactation (>7 days to weaning)
(CT) g/d = 199 (±22.8) + 78 (±15.7) × BW (kg); R^2^ = 0.16; *p* < 0.0001
(NC) g/d = 158 (±23.0) + 112 (±16.2) × BW (kg); R^2^ = 0.24; *p* < 0.0001
Weaned piglets (5 weeks after weaning)
(CT) g/d = 238 (±19.6) + 55 (±13.4) × BW (kg); R^2^ = 0.16; *p* < 0.0001
(NC) g/d = 232 (±18.8) + 63 (±13.2) × (BW (kg); R^2^ = 0.24; *p* < 0.0001
Growing-Finishing (until slaughter):
(CT) g/d = 768 (±21.7) + 86 (±14.9) × (BW (kg); R^2^ = 0.16; *p* < 0.0001
(NC) g/d = 756 (±20.0) + 97 (±14.4) × BW (kg); R^2^ = 0.24; *p* < 0.0001

**Table 3 animals-12-02936-t003:** Effect of parity (P), neonatal care treatment (T), birth weight (BW), and sex (S) on carcass characteristics and splitting yield (covariable carcass weight) (CT: control; NC: neonatal care; CM: castrated male; F: female; low BW: weight equal to or less than 1.1 kg; normal BW: weight over 1.1 kg).

		Parity (P)		Treatment (T)		Birth Weight (BW)		Sex (S)		SD					*p*-Value				
		Gilt	Sow	CT	NC	Low	Normal	CM	F		P	T	BW	S	PxT	PxBW	TxBW	PxTxBW	Cov
Carcass																			
	% Lean	60.01	59.76	59.74	60.02	59.35	60.41	59.13	60.63	2.66	0.401	0.336	0.002	0.001	0.632	0.897	0.836	0.818	0.001
	SC fat (P2, mm)	16.46	16.77	16.85	16.39	17.11	16.13	17.40	15.83	3.24	0.351	0.156	0.003	0.001	0.498	0.681	0.560	0.561	0.001
	SC fat (Ham, mm)	12.09	12.29	12.32	12.06	12.45	11.93	12.79	11.60	2.59	0.427	0.304	0.043	0.001	0.647	0.364	0.575	0.585	0.001
	Cascass weight loss	0.18	0.17	0.14	0.21	0.11	0.24	0.11	0.23	0.01	0.853	0.368	0.068	0.096	0.487	0.430	0.091	0.726	0.001
Splitting yield																			
	Loin weight (kg)	3.27	3.27	3.30	3.24	3.19	3.35	3.18	3.37	0.57	0.996	0.532	0.087	0.047	0.857	0.155	0.519	0.841	0.001
	Ham weight (kg)	12.85	12.82	12.83	12.84	12.73	12.94	12.69	12.98	1.28	0.635	0.851	0.002	0.001	0.405	0.781	0.316	0.212	0.001
	Hams muscle (kg)	7.36	7.37	7.31	7.42	7.24	7.49	7.14	7.59	0.86	0.930	0.142	0.003	0.001	0.530	0.886	0.491	0.203	0.001

**Table 4 animals-12-02936-t004:** Effect of parity (P), neonatal care treatment (T), birth weight (BW), and sex (S) on selected meat quality characteristics (covariable carcass weight) (CT: control; NC: neonatal care; CM: castrated male; F: female; low BW: weight equal to or less than 1.1 kg; normal BW: weight over 1.1 kg).

		Parity (P)		Treatment (T)		Birth Weight (BW)		Sex (S)		SD					*p*-Value				
		Gilt	Sow	CT	NC	Low	Normal	CM	F		P	T	BW	S	PxT	PxBW	TxBW	PxTxBW	Cov
Meat quality																			
	pHu	5.55	5.49	5.51	5.53	5.54	5.51	5.55	5.50	0.17	0.044	0.420	0.368	0.128	0.869	0.531	0.784	0.469	0.105
	EC	3.60	3.68	3.60	3.68	3.60	3.68	3.50	3.77	1.32	0.746	0.733	0.759	0.270	0.241	0.853	0.025	0.970	0.684
	Drip loss	0.03	0.03	0.03	0.03	0.02	0.03	0.02	0.03	0.02	0.312	0.885	0.082	0.006	0.532	0.829	0.956	0.681	0.007
	Soluble protein	119.15	111.05	113.16	117.04	116.62	113.58	117.45	112.75	38.12	0.229	0.569	0.665	0.488	0.522	0.044	0.776	0.086	0.672
Intramuscular fat																			
	LD	2.17	2.03	2.13	2.07	2.29	1.91	2.38	1.82	0.85	0.341	0.667	0.010	0.001	0.348	0.433	0.617	0.888	0.002
	BF	3.09	2.81	2.96	2.94	3.30	2.60	3.25	2.65	1.06	0.108	0.886	0.001	0.002	0.914	0.175	0.271	0.267	0.001
	SM	3.61	3.16	3.49	3.28	3.56	3.21	3.60	3.16	0.97	0.005	0.200	0.033	0.007	0.094	0.091	0.583	0.563	0.015
Moisture																			
	LD	74.62	74.53	74.47	74.68	74.35	74.80	74.29	74.86	0.90	0.492	0.161	0.003	0.001	0.922	0.597	0.620	0.835	0.001
	BF	75.13	75.22	75.19	75.16	74.83	75.52	74.73	75.62	1.29	0.663	0.915	0.001	0.001	0.132	0.998	0.345	0.328	0.001
	SM	73.95	74.20	73.94	74.21	73.93	74.23	73.81	74.35	0.95	0.128	0.095	0.074	0.001	0.392	0.766	0.235	0.715	0.002
a value																			
	LD	10.76	10.43	10.69	10.51	10.57	10.62	10.63	10.57	0.79	0.024	0.214	0.741	0.670	0.535	0.839	0.953	0.249	0.798
	BF	15.35	14.72	15.30	14.77	15.01	15.06	15.43	14.64	1.83	0.043	0.097	0.875	0.014	0.695	0.072	0.048	0.342	0.977
	SM	15.53	14.86	15.41	14.99	15.29	15.11	15.49	14.90	1.47	0.009	0.098	0.500	0.022	0.818	0.194	0.246	0.267	0.344
b value																			
	LD	14.25	14.38	14.27	14.36	14.30	14.34	14.24	14.40	0.67	0.264	0.405	0.731	0.151	0.218	0.252	0.773	0.756	0.721
	BF	15.24	15.22	15.26	15.19	15.25	15.20	15.50	14.96	1.53	0.932	0.809	0.833	0.045	0.121	0.116	0.632	0.977	0.808
	SM	16.05	16.08	16.13	16.00	16.09	16.04	16.26	15.87	1.31	0.906	0.593	0.827	0.097	0.705	0.135	0.282	0.237	0.758
L value																			
	LD	58.94	59.60	59.33	59.20	59.24	59.29	58.81	59.72	2.49	0.137	0.770	0.917	0.041	0.341	0.264	0.925	0.513	0.223
	BF	52.77	53.20	52.65	53.32	52.81	53.16	53.21	52.76	3.10	0.451	0.237	0.544	0.437	0.117	0.110	0.446	0.645	0.985
	SM	54.99	55.58	55.33	55.24	55.10	55.46	55.38	55.19	2.87	0.271	0.858	0.513	0.723	0.744	0.197	0.149	0.503	0.180

**Table 5 animals-12-02936-t005:** Effect of parity (P), neonatal care treatment (T), birth weight (BW) and sex (S) on fatty acid composition of intramuscular lipids (covariable carcass weight) (CT: control; NC: neonatal care; CM: castrated male; F: female; low BW: weight equal to or less than 1.1 kg; normal BW: weight over 1.1 kg).

	Parity (P)		Treatment (T)		Birth Weight (BW)		Sex (S)		SD					*p*-Value				
	Gilt	Sow	CT	NC	Low	Normal	CM	F		P	T	BW	S	PxT	PxBW	TxBW	PxTxBW	Cov
C14:0	1.50	1.29	1.47	1.32	1.49	1.31	1.27	1.52	0.14	0.235	0.732	0.814	0.171	0.988	0.522	0.396	0.025	0.001
C16:0	23.84	23.47	23.55	23.75	23.87	23.43	23.96	23.34	1.17	0.060	0.321	0.032	0.002	0.921	0.539	0.931	0.716	0.002
C16:1, n-9	0.24	0.24	0.24	0.24	0.23	0.25	0.23	0.25	0.04	0.876	0.579	0.006	0.041	0.808	0.947	0.943	0.953	
C16:1 n-7	3.42	3.28	3.23	3.46	3.44	3.26	3.41	3.28	0.56	0.109	0.015	0.049	0.160	0.974	0.157	0.678	0.650	0.001
C18:0	11.86	11.92	11.97	11.81	11.84	11.94	12.03	11.75	0.90	0.678	0.302	0.513	0.081	0.856	0.515	0.878	0.476	0.249
C18:1 n-9	39.79	40.07	39.80	40.05	40.66	39.20	40.31	39.54	2.18	0.454	0.510	0.001	0.044	0.335	0.245	0.591	0.644	0.002
C18:1 n-7	3.58	3.47	3.44	3.62	3.59	3.46	3.54	3.51	0.43	0.156	0.020	0.109	0.702	0.782	0.398	0.959	0.611	0.041
C18:2 n-6	10.34	10.38	10.57	10.15	9.70	11.02	9.77	10.95	2.24	0.897	0.257	0.002	0.002	0.552	0.322	0.634	0.997	0.001
C18:3 n-3	0.35	0.35	0.36	0.35	0.34	0.37	0.34	0.36	0.05	0.906	0.260	0.002	0.016	0.665	0.771	0.555	0.419	0.015
SFA	37.54	37.19	37.35	37.38	37.53	37.20	37.84	36.89	1.76	0.255	0.915	0.297	0.003	0.998	0.959	0.907	0.430	0.094
MUFA	47.96	48.02	47.69	48.30	48.86	47.12	48.45	47.53	2.73	0.910	0.196	0.002	0.052	0.425	0.200	0.613	0.592	0.001
PUFA	14.52	14.76	14.97	14.31	13.60	15.67	13.71	15.57	3.56	0.671	0.266	0.002	0.002	0.536	0.299	0.681	0.989	0.002
UI	84.52	85.46	85.55	84.43	83.06	86.92	82.91	87.07	7.50	0.440	0.367	0.003	0.001	0.582	0.372	0.752	0.776	0.002

## Data Availability

Not applicable.

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
