# Peer review of "Short- and Long-Term Effects of Birth Weight and Neonatal Care in Pigs"

_animals, 2022, doi:10.3390/ani12212936_

Round 1
Reviewer 1 Report
The article titled short- and long-term effects of birth weight and neonatal care in pigs is an interesting read and focuses not only on the immediate effects of intervention within the farrowing environment but also the end product. This is critical and missed in a lot of studies. The observation of long-term effects of neonatal care on metabolism and thus potentially carcass and eating quality provides a new perspective on the value of interventions rather than just mortality.
General:
Overall, an interesting topic however the grammar and language use made it difficult to read. I suggest a native English speaker read over the manuscript.
The page numbers are out of order, page 9-11 have no numbers and from 12 onwards they are the wrong page numbers.
Your references are out of order. e.g., Christison et al is referenced as 29 but in your list is 28. You also have SAS referenced between 40 and 41 however I believe this can be referred to in text without a reference list record.
I also heavily suggest that you review the references used. There are only 10 from the last 7 years published with a large majority close to 20+ years old. There have been many new neonatal cares and or mortality and meat science publications in the last 5 years that will enhance your manuscript.
Consistency: Throughout your manuscript you switch between low body weight and birth weight as terms as well as descriptions for BW. Birth weight in these types of papers is most commonly accepted.
Abstract:
Line 33: Insert BW in “low piglet”
L34-35: Remove or replace mainly at first week of live. “A larger difference in mortality was observed within the first week post birth.” And add P value.
The abstract in whole is missing P-Values or other numerical evidence. Add to strengthen descriptions.
Introduction:
The Introduction is very brief, for an area quite heavily researched there is definitely more that could be discussed. Reviewing the literature will help to improve this section
L56: change <1-1.1kg to either the actual bottom number -1.1kg or just <1.1Kg.
L56: rearrange sentence to the proportion of small piglets () in litters with more than 15 piglets is above 15-20%.
L58: what strategies? Elaborate on some used and evidence for why you picked the ones you did
L60: change productive circumstances to “within production”
Materials and Methods:
L88: clarification of parity profiles will be made clearer if written P2 = 4, P3 = 16 etc.
L89: which group does the second parity sows belong to? Suggest labelling groups e.g., Parity group (PG) 1 = gilts, PG2 = parity 2 – 8 sows)
L108-9: unclear sentence. Possibly an alternative “The daily work routine was kept consistent across the farrowing systems to reduce source of variation.”
Statistical Analysis:
Did you test for the litter size effect on mortality? You cross fostered to the teat availability but not all sows have the same number of teats. Reared as well as birth litter size has shown to massively influence survival.
Results:
Check use of tense. Results should be written in past tense. Was not.
L215: P values should be reported as P= … not < or > unless they are reported as <0.001. Check all other uses.
L242: Referring to table 2? your graph does not show several stages of growth, but your table does.
L243: Remove “however it is interesting to note”. Results are just observations not explanations. “A tendency to ….
L244: two brackets next to each other can be removed and replaced with, e.g., P = 0.74; Figure 2)
L252: P-value not the same as reported in the table. P =0.091
L251 and 261: Since you saw a difference between sexes in fat did you consider doing a sex* Bw interaction? This may account for some of the variation.
L263-268: no pvalues included. Please add.
Tables and Figures:
All table and Figure legends need more details. Explain abbreviations e.g., Ctr and NC, and categories such as Low and Normal weights, in caption. See other papers for suggestions.
Figures need better identification of which graph is which. Add A and B to top left of graphs
Add y axis labels and error bars
Figure 3 is not referred to in text.
Discussion:
Use of subheadings should be either throughout the results as well for continuity or not at all.
Lines290-293, L302-306 and L308-311 are almost word for word from the results section. You need to change these so that they are not exactly the same either here or in the results section.
L293: for better flow suggest directly comparing to mortality reported in the literature ... Our results are in agreement with previously reported data…. That strengthens the argument that bw is a main contributing factor to neonatal mortality.
L297: this should be expanded on. Does this paper reporting a higher mortality at a lower threshold suggest other factors involved or that a larger sample size might show a that this threshold is actually lower?
L302: remove “considered” as it is well documented.
L307: you should not be referring to figures or tables in the discussion.
L314: Christion is not 29
L317: Avoid using the same sentence starters e.g. Moreover, within the same paragraph.
L327: remove brackets or all together.
L331: “of” P x BW
L340-344: good justification but need to refer to other literature. There are some papers which show that smaller piglets have less of a capacity for compensatory growth or longer to market weights. These would support this argument.
L358-9: Is the sex effect on intramuscular fat and moisture content expected? What is reported? Needs to compare to literature
L367-8: You say its difficult to explain but is there anything that shows a similar pattern with ph or other measures? Even in other species?
L386: Replace “On the one hand”. You do not directly compare to the alternative within the same sentence or not, so this phrase is not appropriate.
L397: If above is removed then change this to “Alternatively”
Conclusion:
L408: add either normal or low in front of BW
Reviewer 2 Report
The aim of this research was to evaluate the influence of piglet body weight at birth and individual neonatal care of newborn piglets on survival, growth, carcass and meat quality. It is known that sire influence the vitality and survival of piglets, but also the growth and quality of carcass, it is necessary to include the boar-sire in the model as a random effect. Only then, it will be possible to look more objectively at the effects of other factors and "make a conclusions.
Author Response
The comments are below.

Reviewer 3 Report
The authors aimed to study the effect of BW at birth and individual neonatal care provided to neonatal piglets according to sow parity and sex on mortality along lactation and long term effect on growth, and carcass and meat characteristics.
This manuscript can be accepted after major revision.
1. Line 51: “piglets die during the first days of life”. Please give some causes as an example.
2. Line 198: “p-value < 0.10 was considered a trend.” This is only the case when the experiment is really unknown effect (new approach), so this is not true for this stud. Please interpret only statistically significant (p<0.05) cases.
3. Line 202: “Fix effect”à” Fix effects”
4. What is the multiple comparison test “such as Duncan”?
5. It is not clear whether the researchers did homogeneity and normality tests to understand the data is suits for analysis of variance. Should we trust the results?

Author Response
The comments are below (attatched document).

Round 2
Reviewer 1 Report
An overall good manuscript and important area for publication.
A much-improved manuscript with greater flow, clarity and reference to other relevant papers. Still a fair number of grammatical and word choice suggestions below to allow for further clarification along with a few critical points.
Main points:
Make sure to mention the factors like litter size that you tested in your models even if they were not included in the final model
Check for consistency of abbreviations and terms
Discussion section does not include results. In the sense that figures, tables, references to them and statistics (p-values and means) should not be reiterated in the discussion section as that is why the results section is there. The discussion should focus on what this means then. “An increase in … similar to that in (other paper) lends strength to the idea that piglet intervention is beneficial to some but not all. …”
I understand the figure 3 lends to the discussion as does all the other results. Since this is true it should be in the results. Nothing not mentioned in the results can be discussed in the discussion section.
Minor:
L20: don’t use BW in simple summary as you have not defined it yet.
L20 remove neonatal once, does not need to be restated twice.
. L22 and 23: neonatal cares. remove the s.
L27: BW at birth is repetitive. removed at birth as it is birth weight. same as above for removal of neonatal.
L27-28. Mortality along lactation. replace with preweaning mortality. it is a standard phrase used in pig production. Check throughout manuscript and replace where appropriate
L29: change to; Litters from seventy-one.... were included in the trial. Half of each litter did not receive.... and the remaining half received ...
L32 When two brackets) (you put; between.
L33: change being to with.
L34-35: Change to but a significant effect was observed in low BW piglets who died of starvation.
L40: low to lower than normal…
L41: and the opposite.
L59-63: too long of a sentence. Stop after performance of neonatal piglets. Additionally, management practices in early lactation including…. Have been suggested.
L64: The combination.
L96: closing bracket missing.
L120: change to; and were returned to the same location within the crate in which they were removed from.
Statistical Analysis:
I would mention then that Litter size was tested but was found non-significant. Otherwise, it would be assumed you did not try to fit it.
Further did you account for the pen effect at any of the production stages on growth and carcass quality? This has been shown in the past to have significant impact on growth (temperature, airflow, water access/pressure). This is often minimised by the design of the housing but is still tested in the modelling.
L223 low BW piglets
L224: mortality was especially concentrated within the first week of life…
L225: remove double.
And start No effect of P a line below. It is confusing with the sentence before.
All Tables: try to keep traits/categories at the same decimal number 2.13 vs 3.0
Table 1:
· replace, with.
· keep P values to the same number of decimal points
· Notes? Incorporate into the table heading. Footnotes are more if you are going to refer to in the table with symbols * etc.
L245-256: A lot of information which is hard to read. I would suggest it being put in a table format to make it easier to follow.
L257: Do you need the figure? Does it add anything more when you have the equations there? Or is there a way to incorporate the two.
L257-259: this is not a result but an explanation. Thus indication… does not belong in the results section. Remove.
Discussion:
You should not be referring to any figures/tables or numbers like 32 vs 10% of your own study. This is what the results section is for. You assume the reader has read to this point. You should only report other studies values if appropriate in discussion of the implications of your outcomes.
Remove figure 3 from the discussion. If it truly adds to the discussion, then it needs to be in the results section.
L269: remove also in this case and change to slopes.
L330: change birth weight to BW (be consistent)
L270: change question the potential interest; to, it may impact the outcomes of.
L371 double space between apparently and may
L401: you responded that there is no bibliography on this subject... Mention it in text. There is not current research in this specific area…
Conclusion:
A nice summary of your main results. I would add a sentence at the end to say what is the implication for pig production and or research…
L448: you did not measure vitality specifically so I would not mention it here. You use other phrases like impaired function, use those to be consistent.
L450 extra space and.
Reviewer 2 Report
Thank you for your reply. It is very unusual to keep data on the parity of sows, the litter, the individual body weight of the piglets, and the sire of the litter is not known. I would like to mention that 71 female (gilts and sows) were included in the experiment. Considering the concept and traits that were the subject of the research, I am not sure of the validity of the obtained results. For this reason, I cannot recommend publication of this manuscript.
